# Inflammasome Coordinates Senescent Chronic Wound Induced by *Thalassophryne nattereri* Venom

**DOI:** 10.3390/ijms24098453

**Published:** 2023-05-08

**Authors:** Carla Lima, Aline Ingrid Andrade-Barros, Fabiana Franco Carvalho, Maria Alice Pimentel Falcão, Monica Lopes-Ferreira

**Affiliations:** Immunoregulation Unit of the Laboratory of Applied Toxinology (CETICs/FAPESP), Butantan Institute, São Paulo 05503-009, Brazil

**Keywords:** *Thalassophryne nattereri*, necrosis, senescence, wound healing, repair, innate cells, neutrophils, M1/M2 macrophages, IL-1α/β, caspase-1/-11, inflammasome complex

## Abstract

*Thalassophryne nattereri* toadfish (niquim) envenomation, common in the hands and feet of bathers and fishermen in the north and northeast regions of Brazil, is characterized by local symptoms such as immediate edema and intense pain. These symptoms progress to necrosis that lasts for an extended period of time, with delayed healing. Wound healing is a complex process characterized by the interdependent role of keratinocytes, fibroblasts, and endothelial and innate cells such as neutrophils and macrophages. Macrophages and neutrophils are actively recruited to clear debris during the inflammatory phase of wound repair, promoting the production of pro-inflammatory mediators, and in the late stage, macrophages promote tissue repair. Our hypothesis is that injury caused by *T. nattereri* venom (V*Tn*) leads to senescent wounds. In this study, we provide valuable information about the mechanism(s) behind the dysregulated inflammation in wound healing induced by V*Tn*. We demonstrate in mouse paws injected with the venom the installation of γH2AX/p16^Ink4a^-dependent senescence with persistent neutrophilic inflammation in the proliferation and remodeling phases. V*Tn* induced an imbalance of M1/M2 macrophages by maintaining a high number of TNF-α-producing M1 macrophages in the wound but without the ability to eliminate the persistent neutrophils. Chronic neutrophilic inflammation and senescence were mediated by cytokines such as IL-1α and IL-1β in a caspase-1- and caspase-11-dependent manner. In addition, previous blocking with anti-IL-1α and anti-IL-β neutralizing antibodies and caspase-1 (Ac YVAD-CMK) and caspase-11 (Wedelolactone) inhibitors was essential to control the pro-inflammatory activity of M1 macrophages induced by V*Tn* injection, skewing towards an anti-inflammatory state, and was sufficient to block neutrophil recruitment and senescence.

## 1. Introduction

Wound healing is a complex process of tissue architecture repair characterized by the differentiation, proliferation, and activation of keratinocytes, fibroblasts, and endothelial cells which are interdependent on the infiltration of innate cells such as neutrophils and macrophages, on the release of pro-inflammatory cytokines, and on growth factors. The active cooperation between these actors promotes re-epithelialization, angiogenesis, and fibroplasia [1]. However, a dysregulated immune system during the wound healing process leads to persistent inflammation and delayed healing, ultimately resulting in chronic wounds. Chronic wound healing is characterized by the prolonged presence of senescent cells and myeloid cell populations, such as macrophages, neutrophils, and monocytes in the late stage of inflammation [2].

Persistent inflammation in chronic wounds is characterized by an abundance of pro-inflammatory macrophages producing tumor necrosis factor-α (TNF-α) and interleukin-1β (IL-1β) with limited ability to eliminate dead neutrophils [3]. Furthermore, chronic wound macrophages release several matrix metalloproteinases (MMP) such as MMP-2 and MMP-9, which prevent the initiation of the proliferative phase of wound healing. As such, novel ways to target inflammatory cell infiltration through the manipulation of key signaling pathways appear to be viable strategies for improving outcomes in chronic wounds.

*Thalassophryne nattereri* toadfish (niquim) is responsible for many accidents in the north and northeast regions of Brazil [4,5]. Accidents with the fish involving fishermen and bathers are more frequent due to the habit of the fish remaining hidden under the sand, between stones or seaweed, and covered by mud in relatively dark places. *T. nattereri* envenomation is more common in the hands and feet, where the spine penetrates human tissue and the integumentary lining surrounding the gland presses the venom through the duct that is injected into the victim through the canalicular spine.

One of the main local symptoms resulting from envenomation is pain immediately after the accident, which persists for more than 24 h and, according to the description of the victims, is of great intensity. Erythema and edema are also noticed immediately with the formation of exudates. Injury can persist, and wounds are not healed in a timely fashion, remaining for an extended period of time [6,7]. Symptoms are aggravated by a lack of specific therapy [8].

An ambiguous role for V*Tn* in lesion induction has been reported. On the one hand, V*Tn* modulates the levels of stress in the microvasculature that triggers the production of cytokines and other pro-inflammatory factors [9]. On the other hand, it delays the removal of necrotic material in the tissue [10], disorganizes the extracellular matrix (ECM), decreases the content of collagen fibers during the proliferative phase dependent on the action of MMP-2 and MMP-9 [11], and hinders the correct infiltration of inflammatory cells. Intravital microscopy confirmed the induction of ischemic damage in the microcirculation, including thrombus formation followed by complete venular and transient arteriolar stasis [12].

The natural progression through the stages of wound healing necessary to ensure adequate repair of the epidermis and dermal architecture in V*Tn* envenomation sounds inadequate, suggestive of a senescent wound. Recently, a large amount of evidence has proven that cellular senescence has a crucial influence on chronic non-healing wounds [13]. Cellular senescence is a process resulting from the accumulation of DNA damage in which cells cease dividing and undergo profound chromatin and secretome changes [14]. Recently, Mahmoudi et al. (2019) [15] found that senescent fibroblasts affect the efficiency of cell reprogramming and wound healing rates, and their secretion of inflammatory cytokines (TNF-α, IL-1β e IL-6) is a critical contributor.

Senescence leads to increased platelet aggregation after injury and decreased vascular permeability, resulting in the inability of macrophages to infiltrate in a timely manner [16]. When recruited, macrophages have reduced phagocytic capacity and produce fewer growth factors [17]. In addition, senescence decreases the capacity for migration and proliferation of keratinocytes and fibroblasts. Senescent fibroblasts also secrete excess matrix MMPs that cause changes in ECM composition [18]. Finally, there is a significant reduction in the revascularization of chronic wounds on which the wound-healing process is intimately dependent [19].

Inflammasome sensors can assemble multimolecular protein complexes in response to various activators, leading to the activation of inflammatory caspases and subsequent maturation of IL-1 family cytokines. Recently, a new and critical role of the inflammasome in cellular senescence has been described [20]. Fernandez-Duran et al. (2022) [21] demonstrated that caspase-4 activation and non-canonical inflammasome assembly by cytoplasmic LPS triggers a senescence phenotype, contributing critically to the establishment of the senescence-associated secretory phenotype (SASP) and the reinforcement of the cell cycle arrest program.

Our hypothesis is that ischemic injury provoked by V*Tn* leads to the induction of cellular stress with morphological and transcriptional changes in the damaged tissue microenvironment that, consequently, can force cell arrest through senescence, which leads to chronic wounds. In this work, we evaluated whether the injury caused by the venom of *Thalassophryne nattereri* is accompanied by senescence and whether its modulation by the inflammasome complex modifies the activity of the cells involved in the outcome of the injury.

## 2. Results and Discussion

### 2.1. T. nattereri Venom Induces Long-Term Necrosis

Typical wound healing is traditionally divided into four sequential stages: hemostasis, which lasts from minutes to hours after tissue damage; acute inflammation, which lasts from one to three days; proliferation, which usually lasts from a few days to a month; and, finally, tissue remodeling, which involves keratinocytes, fibroblasts, macrophages, and endothelial cells to scar formation [2].

The injury caused by the venom of *T. nattereri* in mice is of an ischemic and chronic nature, characterized by a rapid increase in serum levels of creatine kinase (CK) concomitant with a reduction in its activity in the gastrocnemius muscle, thrombosis in venules and veins, congestion and vascular stasis, death of myoblasts, and delay in the removal of necrotic material that remains for more than seven days after envenomation [10,12].

In this work, we reproduced the lesion in the footpad of BALB/c mice induced by the injection of 10 μg of V*Tn* and evaluated the kinetics of the wound-healing process. The use of the in vivo model recapitulates accidents in humans and makes it possible to see the interrelationship of the various cells and mediators involved in inflammation, chronic wound, and senescence, such as epithelial and endothelial cells, senescent cells, fibroblasts, and keratinocytes, as well as myeloid cell populations, such as macrophages and neutrophils.

To evaluate the kinetics of V*Tn*-injured mice, we characterized the morphological changes, especially looking for a thickness of the dermis and epidermis, leukocyte infiltration, and tissue disruption. First, in the microphotographs in Figure 1A, we observed examples of the paws of mice from the control-group on the left and from the V*Tn*-injected group at 35 d (wound remodeling phase) on the right, where permanent injury with scab formation was observed in the heavily edematous paw of V*Tn*-injected group.

In Figure 1B, histological examination of H&E-stained paraffin foot pad sections from the control-group of mice reveals normal cell architecture with a baseline number of infiltrating cells. On the contrary, the V*Tn*-injected group exhibited paw necrosis identified by the disruption of tissue architecture with ghost cells (green asterisk and green arrow) and fewer discernible nuclei, mainly at 4 h, 1 d, 15 d, and 21 d compared to the control-group of mice. Also, the formation of new dermis and epidermis under the necrotic tissue until 35 d was visible. Dermal thickening was evident immediately after 4 h of injection and at 1 d, with an increase at 3 d. Between 7 and 21 days, the thickening decreased but remained high at 35 d after the injection (Figure 2A). The peak of the epidermal thickness was seen after 4 h of the injection, successively declined until 21 d, and increased again at 35 d of the injection (Figure 2B) showing dermis and epidermis swelling.

Histologically, the infiltration of leukocytes showed a peak of cells at 4 h after V*Tn* injection, followed by a slight decrease at 1 d, but high compared to the normal number found in control mice. After three and seven days, there was a resurgence in the influx of cells, which progressively decreased between 15 and 35 days but was still high relative to the control mice (Figure 2C).

During the early stages of wound healing, neutrophils [22] and pro-inflammatory macrophages [23] are the primary immune cells evidenced at the site. In Figure 3, an examination of sections of the paws of control mice stained with anti-mouse MPO antibodies showed an absence of MPO-positive neutrophils. MPO is a lysosomal enzyme released from the azurophilic granules of activated neutrophils which impedes epithelial migration and wound closure in vivo by inhibiting epithelial cell proliferation [24].

When the analysis was performed on the paws of mice injected with V*Tn*, a large accumulation of MPO-positive neutrophils in the epidermis was evident mainly in the hemostasis and inflammatory phases of the lesion, which remained high during the remodeling phase (Figure 3). We quantified a large influx of neutrophils (137 ± 13) at 4 h, which decreased at 24 h (95 ± 13), at 3 d (18 ± 1), and at 7 d (24 ± 1). However, neutrophilia reappears in the lesion at the proliferative phase, 15 d (66 ± 8) and 21 d (62 ± 6), and remains at moderate levels for up to 35 days (63 ± 7) (Figure 4).

Wounds that fail to promote healing are associated with excessive recruitment and retention of neutrophils at the injury site. The potent functions of neutrophils, such as the production of reactive oxygen species (ROS), release of cytotoxic granular contents (proteases such as elastase and MMP-8, MMP-9, and MPO), and formation of extracellular neutrophil traps (NETs), can also cause collateral tissue damage [22] and the emergence of chronic wounds [25].

Our data showing the permanence of neutrophils in the wound remodeling phase concomitant with thickening of the dermis and epidermis is indicative of a chronic inflammatory response and a failure in the macrophage-mediated local phagocytosis process, crucial for wound resolution.

### 2.2. The Venom of T. nattereri Induces Senescent Wound

Cellular senescence is characterized by several unique features, including, in some cell types, a generalized change in chromatin modification, known as senescence-associated heterochromatin foci (SAHF). It is an important additional contributor to lesion chronification with ulcer formation and cell transformation [26,27].

In order to define whether V*Tn*-induced injury was associated with an increased burden of senescent cells, we initially examined a highly sensitive marker of growth arrest and cellular senescence, γH2AX. The carboxy-terminal Serine 139 of H2AX is phosphorylated within 1 to 3 min after damage, and the number of H2AX molecules phosphorylated increases linearly with the severity of the damage [28]. The presence of γH2AX-foci within a population of hematoxylin-counterstained cells was recognized by the formation of a homogeneous brown precipitate partially or entirely covering the cells, which were easily distinguished by their blue-colored nuclei.

Histological examination (Figure 5) and parallel quantification (Figure 6A) show that sections of the paws from control mice stained with anti-γH2AX have a scarce number of positive foci. We found that γH2AX-positive foci were significantly increased in V*Tn* wounds at 4 h (106 ± 15 compared with 22 ± 1.1 of the control-group at 0 h) which progressively increased to 35 d (385 ± 19.36) randomly distributed across all cell types in the wound.

These data suggest that senescence in the wound environment is probably not limited to fibroblasts and macrophages. Corroborating findings show that other wound cells, including keratinocytes [29] and endothelial cells [30], are capable of undergoing senescence in response to inflammatory cues.

Cyclin-dependent kinase (CDK) inhibitor p16^INK4a^ plays multiple biologic functions, including the inhibition of cell cycle progression, the modulation of DNA damage-induced apoptosis, and the induction of senescence. Human fibroblasts overexpress p16^Ink4a^ and undergo senescent growth arrest [31] through p38-MAPK-mediated mitochondrial dysfunction and reactive oxygen species (ROS) production [32].

Next, we found diffuse and strong overexpression of p16^INK4a^ positive foci in wound tissues of mice injected with V*Tn***.** Foci of immunoperoxidase staining for p16^INK4a^ was observed at 1 d (199 ± 19 compared with 16 ± 1.1 of the control-group at 0 h) and progressively reached high levels up to 35 d (279 ± 19, Figure 6B).

Our data show a direct correlation between the accumulation of senescent markers and the impaired ability of paw tissue to resolve and repair inflammation. Senescence can consequently generate changes in the phenotype and activity of immune cells at the site of injury, leading to chronic low-grade inflammation.

### 2.3. VTn-Induced Wound Is Associated with IL-1α and IL-1β via Activation of Inflammatory Caspases

Unlike IL-1β, whose expression is generally inducible, IL-1α is reported to be constitutively expressed at low levels in various cell types such as fibroblasts and keratinocytes. Stimulated neutrophils and macrophages are recognized as important sources of both cytokines [33,34,35]. While IL-1β (activated via inflammatory caspases) has cell-autonomous functions and enhances its own senescence condition, the capacity of membrane-bound IL-1α to activate IL-1R1 signaling in an intracrine and paracrine manner on surrounding cells reinforces SASP release and senescence [36,37,38].

We next addressed the signaling mechanisms of neutrophilia and senescence in chronic wounds developed in response to V*Tn* in mice that had IL-1α and IL-1β blocked and caspase-1 and -11 inhibited before V*Tn* injection.

We observed that 35 days after venom injection, mice that had IL-1α and IL-1β neutralized showed a drastic reduction in the overall accumulation of leukocytes (Figure 7A, 90% reduction for anti-IL-α and 77% for anti-IL-1β) or in the recruitment of MPO-positive neutrophils to paw wounds (Figure 7B, 95% reduction for both neutralizing Abs treatment) and in the number of γH2AX-positive foci (Figure 7C, 94% reduction for anti-IL1-α and 92% for anti-IL-1β), as can be seen from the images in Figure 8.

Previous studies confirmed that caspase-1 and NLRP3 inflammasome components modulate SASP. As well as senescence-associated macrophages, senescent fibroblasts also show increased expression of NLRP3 and caspase-1 [39], and that caspase-11 non-canonical inflammasome acts in immune-mediated senescent cells in vivo [21,40].

Then, to evaluate the dependence of the canonical and non-canonical inflammasome caspases on the chronic lesion induced by V*Tn*, we inhibited caspase-1 catalytic activity with the pharmacological inhibitor AC-YVAD-CMK or caspase-11 by wedalactone before V*Tn* injection. As shown in Figure 7A, we observed that groups of mice treated with caspase inhibitors showed an almost complete reduction in leukocyte infiltration (caspase-1 inhibitor: 91% and caspase-11 inhibitor: 81%). A reduction of 90% in the infiltration of MPO-positive neutrophils was seen in the wounds of mice treated with caspase-1 inhibitor and 92% for those treated with caspase-11 inhibitor (Figure 7B). Subsequently, the number of γH2AX-positive foci was reduced by 95% after treatment with the caspase-1 inhibitor and by 96% for the caspase-11 inhibitor compared to V*Tn*-induced wounds (Figure 7C). These quantitative changes can be verified in the microphotographs in Figure 8.

These data show that both chronic neutrophilic inflammation and senescence maintained in chronic wounds induced by V*Tn* are mediated by the innate cytokines IL-1α and IL-1β, in a caspase-1 and caspase-11 dependent manner. They also point to a relationship between a γH2AX/p16^Ink4a^-dependent senescence and persistent neutrophilic inflammation.

### 2.4. M1/M2 Macrophage Imbalance in VTn-Induced Chronic Injury Is Dependent on IL-1α and Caspase-1 and Caspase-11

There is evidence for the role of senescent cells and their interaction with macrophages in tissue injury and repair. Indeed, many SASP factors attract and maintain macrophages with a resilient pro-inflammatory phenotype beyond the initial inflammatory phase of wound healing [3,41,42]. Senescent fibroblasts also secrete excess matrix MMPs that cause changes in the ECM composition of the wound, compromising cell migration [15]. Thereby, as macrophages are essential immune cells for the elimination of senescent cells, changes in their appropriate phenotypes lead to high levels of senescence markers and SASPs with sustained inflammation [43,44].

Although an accumulation of evidence challenges the classic M1 and M2 classification, M1-macrophages could be characterized by reduced intracellular glutamine and increased succinate accumulation [45,46] with enhanced glucose flux through glycolysis, which is related to inhibited glutamine-synthesizing enzymes [47]. IL-10-producing M2 is important for the phagocytosis of senescent and apoptotic cells in the acute inflammatory phase of wound healing [48,49], in addition to preventing the production of IL-1β dependent on the activation of NLRP3/Caspase-1 [50]. The efferocytosis of apoptotic neutrophils by M2 macrophages skews toward an anti-inflammatory state with the secretion of resolvins and lipoxins that signal the restoration of vascular integrity and tissue regeneration [51].

Finally, we evaluated whether V*Tn*-induced chronic senescent injury caused changes in the balance of M1/M2 resident macrophages in the footpads of mice injected with V*Tn* at 35 days (the wound remodeling phase). As described in the Figure 9A, CD11b^pos^ macrophages at the R1 gate were sequentially identified in the R2 gate as M1 (F4/80^pos^ CD68^pos^) or M2 (F4/80^pos^ CD206^pos^) by the expression of CD68 (belonging to the *lamp* (lysosomal-associated membrane protein) family of glycoproteins associated with antigen processing or protection of lysosomal membranes against lysosomal hydrolases) or CD206 (a cell-surface protein also known as mannose receptor C type 1-MRC1), respectively.

The frequency of M1 macrophages staining positive for TNF-α, or M2 staining positive for CD36 (a receptor that binds and phagocytoses apoptotic cells and other oxidized lipids), IL-10, or Arginase-1 involved in the wound healing process [52] were investigated as demonstrated in the R3 in the control-group, V*Tn* 35d, anti-IL1a/V*Tn*, casp-1 inhibitor /V*Tn*, and casp-11 inhibitor/V*Tn*.

Interestingly, we observed that in the late remodeling phase, V*Tn* maintained a higher number of TNF-α-producing M1 macrophages in the foot wound (Figure 9B) compared to the control-group. In this period, on the contrary, the number of M2 CD36-positive macrophages with efferocytosis capacity [53] was reduced (Figure 9C). In Figure 9D, we analyzed M2 activity and observed a decrease in the number of IL-10-producing M2 as well as in the number of arginase-1^low^-producing M2. However, V*Tn* maintained an elevated number of arginase-1^high^-producing M2 during this late phase.

These data show us that V*Tn* induces an imbalance of M1/M2 macrophages, maintaining in the reparative-regenerative phase a population of macrophages with a pro-inflammatory M1 activation state, unable to phagocytose apoptotic neutrophils *in situ*, which provides signals for senescence.

As previously demonstrated by Sindrilaru et al. (2011) [54], unrestrained pro-inflammatory M1 macrophages, via enhanced TNF-α and hydroxyl radical release, perpetuate inflammation and induce a p16^INK4a^-dependent senescence program in resident fibroblasts, eventually leading to impaired wound healing. More recently, using a chronic compression injury mouse model, Shimada et al. (2020) [55] demonstrated that M1 macrophage infiltration exacerbates muscle/bone atrophy after peripheral nerve injury, which was abolished by inhibiting local M1 macrophages. On the other hand, using CCL5 inhibitors or CCL5 neutralizing antibodies to induce macrophages to polarize from the M1 to M2 phenotype promotes liver parenchymal regeneration, thereby significantly reducing paracetamol-induced acute liver injury [56].

Interestingly, specific neutralization of IL-1α or inhibition of caspase-1 and caspase-11 activity dramatically blocked the influx of TNF-α-producing M1 to the injured footpad of V*Tn*-injected mice (Figure 9B). In addition, only selective inhibition of caspase-1 (21% increase) and mainly caspase-11 (87% increase) promoted the appearance of CD36-positive M2 macrophages (Figure 9C).

When intracytoplasmic IL-10 or arginase-1 levels were evaluated in M2 macrophages in mice that had IL-1α neutralized, we observed an increase in the number of IL-10 producing- and arginase-1^low^ producing-M2 in relation to the V*Tn*-group but lower than the basal level of the control mice. The neutralization of both caspases’ activity also promoted an increase in the infiltration of IL-10 producing-M2 in relation to the V*Tn*-group but lower than the baseline level of the control mice. No alteration in the high number of arginase-1^high^ producing-M2 was observed after all treatments (Figure 9D).

Our data show that treatment by inhibition of key molecules of the inflammasome complex, such as IL-1α, IL-1β, and caspase-1 and caspase-11, proves to be an important way to control the abnormal recruitment of inflammatory cells, mainly neutrophils, in the late phase of the wound remodeling process. Inhibition of these molecules seems essential to control the pro-inflammatory activity of M1 macrophages, switching to an M2 anti-inflammatory state capable of efferocytosis and, consequently, capable of preventing the γH2AX/p16^INK4a^-dependent program in the wound.

## 3. Materials and Methods

### 3.1. Mice

Female 7–8 weeks old BALB/c wild-type mice were obtained from a colony at Butantan Institute. Mice were maintained in sterile microisolators with sterile rodent feed and acidified water and were housed in positive-pressure air-conditioned units (25 °C, 50% relative humidity) on a 12 h light/dark cycle. The experiments were carried out under the laws of the National Council for Animal Experiment Control (CONCEA) and approved by the Butantan Institute’s Animal Use Ethics Commission (Permit Number: #3778090317 and #1226090317).

### 3.2. Thalassophryne Nattereri Venom

All necessary permits for *Thalassophryne nattereri* capture, conservation, and venom collection were approved by the IBAMA (*Instituto Brasileiro do Meio Ambiente e dos Recursos Naturais Renováveis*, Permit Number: 14.693-1). V*Tn* was obtained from fresh captured specimens at the Mundau Lake in Alagoas, state of Brazil, with a trawl net from the muddy bottom of a lake. Fish were transported to the Immunoregulation Unit of Butantan Institute. V*Tn* was immediately extracted from the openings at the tips of the spines by applying pressure at their bases. After centrifugation, venom was pooled and stored at −80 °C before use. Endotoxin content was evaluated (resulting in a total dose < 0.8 pg/mL LPS) with QCL-1000 chromogenic *Limulus amoebocyte* lysate assay (Bio-Whittaker, Lonza, Australia) according to the manufacturer’s instructions.

### 3.3. Induction of Paw Injury by VTn and Treatments

According to previously published data [7,9], the concentration of 0.33 mg/mL was chosen for inducing necrosis in mice paws. Then, V*Tn* (10 μg of protein) in a fixed volume of 30 μL of sterile PBS was subcutaneously injected into the plantar region of the right hind foot paw of three to five mice per group (V*Tn*-group). Mice injected only with 30 μL of sterile PBS were considered the control-group. For drug treatment experiments, independent groups of mice (three–five mice/group) were treated 30 min before the application of V*Tn* with intraperitoneal (i.p.) injection of 200 μL of monoclonal mouse IgG1 anti-IL-1α (mabg-mIL-1α-5, Invivogen, Toulouse France, anti-IL-1α/V*Tn*-group) or polyclonal goat IgG anti-IL-β (AF1009, anti-IL-1β/V*Tn*-group) neutralizing antibodies, both at 2 μg/mL of caspase-1 inhibitor (Ac YVAD-CMK, 17-8603-78-6, Invivogen–Casp-1 inhib/V*Tn*-group) at 1.76 mg/mL and caspase-11 inhibitor (Wedelolactone, sc-200648, Santa Cruz, Dallas, Texas, USA -Casp-11 inhib/V*Tn*-group) at a dose of 80 μM. An equivalent volume of PBS was similarly injected into mice in the V*Tn* group. All analyses were carried out 4 h, 1 d, 3 d, 7 d, 15 d, 21 d, and 35 d after V*Tn* injection. Experiments were performed independently two times.

### 3.4. Cell Suspension Collection and Phenotypic Analysis by Flow Cytometry

The skin was removed from one hind paw, and the paw was cut into four pieces. Pieces of tissue were incubated in 1 mL RPMI 1640 supplemented with 1 mg/mL of type II collagenase (Roche) and 500 U DNase I (Sigma-Aldrich, Darmstadt, Germany) for 90 min at 37 °C. Digested soft tissue from the paw joint was mechanically disrupted in GentleMacs dissociator (Miltenyi, Paris) and then passed through a 70-μm strainer to obtain a single-cell suspension. The suspension which had the erythrocytes lysed with 0.14 M NH_4_Cl and 17 mM Tris-Cl (pH 7.4), and the total cell suspension was counted and stained for flow cytometry. For labeling molecules exposed on the cell membrane, single-cell suspensions (1 × 10^6^ cells in 100 μL) were first blocked by incubation for 30 min in a solution containing 3% normal mouse serum in ice. Afterward, the suspensions were incubated with specific anti-mouse Abs fluorochromes-conjugated or purified Abs followed by secondary Abs fluorochromes-conjugated purchased from BD Biosciences, R&D Systems, or eBioscience: Anti-mouse CD11b clone M1/70 (FITC, 10-0112-82 or PE, 12-0112-82) and anti-mouse F4/80 (PECy5, 15-4801-82 or clone 6F12, 552958),anti-rat APC (17-4822-82) or FITC (14-4811-85), CD68 purified (IMG80112),anti-mouse FITC (11-4011-85), CD68 PE clone FA11 (12-0681-82), CD36 APC (17-0361-82), CD206 ALEXA FLUOR 488 (polyclonal, FAB2535G), PE (FAB2535P), or APC (FAB2535A) for 30 min on ice. Cells were washed three times in RPMI medium and re-suspended in paraformaldehyde 1% for the cytofluorometric analysis.

For the intracellular labeling of the TNF-α, IL-10, and arginase, the clone MP-6-XT22 (APC, 17-7321-82), the clone JES5-16E3 (7101-85), and the goat polyclonal IgG (NB100-59740) were added and mixed with secondary antibody 12-4822-82. The cells were incubated with a Golgi-plugTM solution containing brefeldrin A (BD Biosciences), fixed, and permeabilized for 20 min at 4 °C with the solution Cytofix/Cytoperm (BD Biosciences). Appropriate isotype controls were used as negative controls and to set the flow cytometer photomultiplier tube voltages. Single-color positive controls were used to adjust instrument compensation settings. Cells were examined for viability by flow cytometry using side/forward scatter characteristics. Data (100,000 events acquired per sample) were acquired using a four-color FACSCalibur flow cytometer equipped with CellQuest software (Becton-Dickinson, San Jose, CA, USA). Analysis was performed by trained staff at the Immunoregulation Unit, Laboratory of Applied Toxinology. Data were recorded as percents of fluorescent positive cells, MFI, or absolute number as appropriate.

### 3.5. Hematoxylin/Eosin Staining

For histological analysis, a group of mice (control, V*Tn,* or treated/V*Tn*) sacrificed after 4 h, 1 d, 3 d, 7 d, 15 d, 21 d, and 35 d had their paws amputated. They were fixed in a 10% formalin solution, then paraffinized, sectioned at 5 μm, and fixed on a slide to receive hematoxylin/eosin (H & E) staining for the evaluation of the inflammatory infiltrate. The images were obtained with the Axio Imager A.1 microscope, Carl-Zeiss, Germany coupled to the Zeiss AxioCam IcC1 camera in 10× or 40× objectives using 1.6 optovar. The cells were counted in tissue by manual scoring on a 400 mm^2^ section per slide and performed using Axio Vision Rel 4.8 software.

Epidermal and dermal thicknesses were measured in photomicrographs of paw sections obtained after hematoxylin and eosin staining by randomly selecting six regions using an image analysis system with a Zeiss AxioCam IcC1 camera in 10× or 40× objectives using 1.6 optovar. The vertical thickness of the whole paw and the thickness of the epidermal layer were defined as the distance from the panniculus carnosus to the stratum corneum and as the distance from the basal layer to the stratum corneum, respectively.

### 3.6. Immunohistochemistry

A group of mice (control, V*Tn,* or treated/V*Tn*) sacrificed after 4 h, 1 d, 3 d, 7 d, 15 d, 21 d, and 35 d had their paws amputated and were fixed in a 10% formalin solution. They were paraffinized, sectioned at 5 μm, and then deparaffinized with xylene, rehydrated with alcohol and PBS (2× for 1 min), and fixed with 3% formaldehyde for 30 min at room temperature. After a new cycle of washes with PBS (3× for 5 min), a 3% hydrogen peroxide solution in PBS was added to block endogenous peroxide. After 10 min of incubation at room temperature, the sections were washed with PBS (3× for 5 min) and incubated with 0.1% trypsin solution added with 0.1% CaCl_2_ for 10 min, followed by the addition of 4 M HCl solution for 15 min at room temperature. The reaction was neutralized by adding PBS Tween 0.1% + BSA 10% for 2 h.

DNA damage detection according to Krishnamurt et al. (2004) [28] was performed by staining for 1 h with the monoclonal mouse IgG2a anti-mouse phosphohistone γH2AX antibody (Ser139, MAB3406, R & D Systems, Minneapolis, MN, USA) or with goat polyclonal anti-mouse p16^ink4a^/cdkn2a (NB100-40911, Novus Biologicals, Centennial, CO, USA) at 10 μg/mL. The detection of neutrophils according to Zenaro et al. (2015) [57] was done by incubation with monoclonal IgM anti-mouse myeloperoxidase antibody (MPO, sc-271881, Santa Cruz) at a 1/500 dilution. Then, the detection antibodies were added: goat anti-mouse IgG HRP (sc-2005, Santa Cruz, Dallas, Texas, USA) or anti-mouse IgM HRP (A8786, Sigma), both at 1/5000 dilution. Controls were performed by omitting primary antibodies from the immunohistochemistry procedure.

After a new wash cycle, a 3′,3′ diaminobenzidine (DAB) chromogenic solution (D4168, Sigma) was added with H_2_O_2_ for 5 min in the dark. Sections were counterstained with hematoxylin to visualize morphology. The images were obtained with the Axio Imager A.1 microscope, Carl-Zeiss, Germany, coupled to the Zeiss AxioCam IcC1 camera in 10×, 40×, or 100× objectives using 1.6 optovar.

Foci of immunoperoxidase staining for γH2AX or p16^INK4a^ in either the nucleus or cytoplasm were counted in tissue by manual scoring on a 400 mm^2^ section per slide and performed using Axio Vision Rel 4.8 software. Discrete foci were counted as individual entities. Foci that appeared to overlap or merge were counted as one entity unless the overlap could be visibly distinguished.

### 3.7. Statistical Analysis

All values were expressed as mean ± SEM. Experiments using three to five mice per group were performed independently two times. Parametric data were evaluated using analysis of variance, followed by the Bonferroni test for multiple comparisons. Non-parametric data were assessed using the Mann-Whitney test. Differences were considered statistically significant at *p* < 0.05 using GraphPad Prism (Graph Pad Software, v6.02, 2013, La Jolla, CA, USA).

## 4. Conclusions

Our data together demonstrate that the injury induced by *Thalassophryne nattereri* venom is characterized by persistent neutrophilia, pro-inflammatory M1 macrophages, and senescent cells characteristic of a non-healing wound. Once in the wound, these senescent cells release large amounts of pro-inflammatory cytokines, such as IL-1α and IL-1β, in an inflammasome-dependent manner that together contribute to extensive degradation of the connective tissue and increase their own condition of senescence. Finally, we describe here that the treatment of the chronic wound induced by V*Tn* targeting the inflammasome pathway, such as neutralization of IL-1α and IL-1β and inhibition of caspases activity, represents other therapeutic means intended for non-healing wounds.

## Figures and Tables

**Figure 1 ijms-24-08453-f001:**
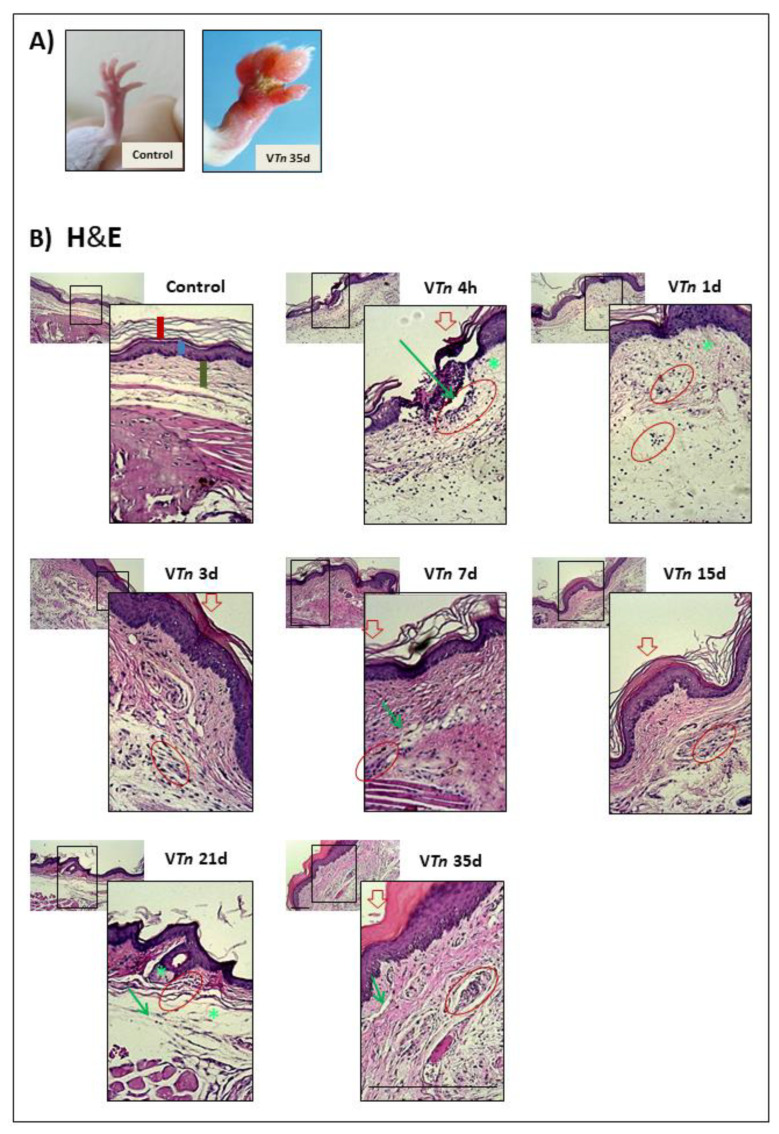
*T. nattereri* venom induces long-term wound in the footpads of mice. Female BALB/c mice were s.c. injected into the intraplantar region with 10 µg of the V*Tn* diluted in 30 µL of sterile PBS, and the control-group was injected only with sterile PBS. (**A**) Examples of the paws of mice from the control-group on the left and from the V*Tn*-injected group at 35 d (wound remodeling phase) on the right, where permanent injury with scab formation was observed in the V*Tn*-injected group. After 4 and 24 h, 3 days, 7, 15, 21, and 35 days of injection, mice had their paws amputated, fixed, paraffinized, and sectioned at 5 μm. Slides were stained with H & E (**B**). Changes in the structure of the layers were indicated by red circles (leukocyte infiltrate), green asterisks, thin green arrows (ghost cells), and closed red arrows (scab formation). In the photomicrograph of the normal paw of the control-group, the vertical red bar indicates the stratum corneum, the vertical blue bar indicates the epidermis and the vertical green bar indicates the dermis. The images were obtained with the Axio Imager A.1 microscope, Carl-Zeiss, Germany, coupled to the Zeiss AxioCam IcC1 camera in 10×, 40×, or 100× objectives using a 1.6 optovar.

**Figure 2 ijms-24-08453-f002:**
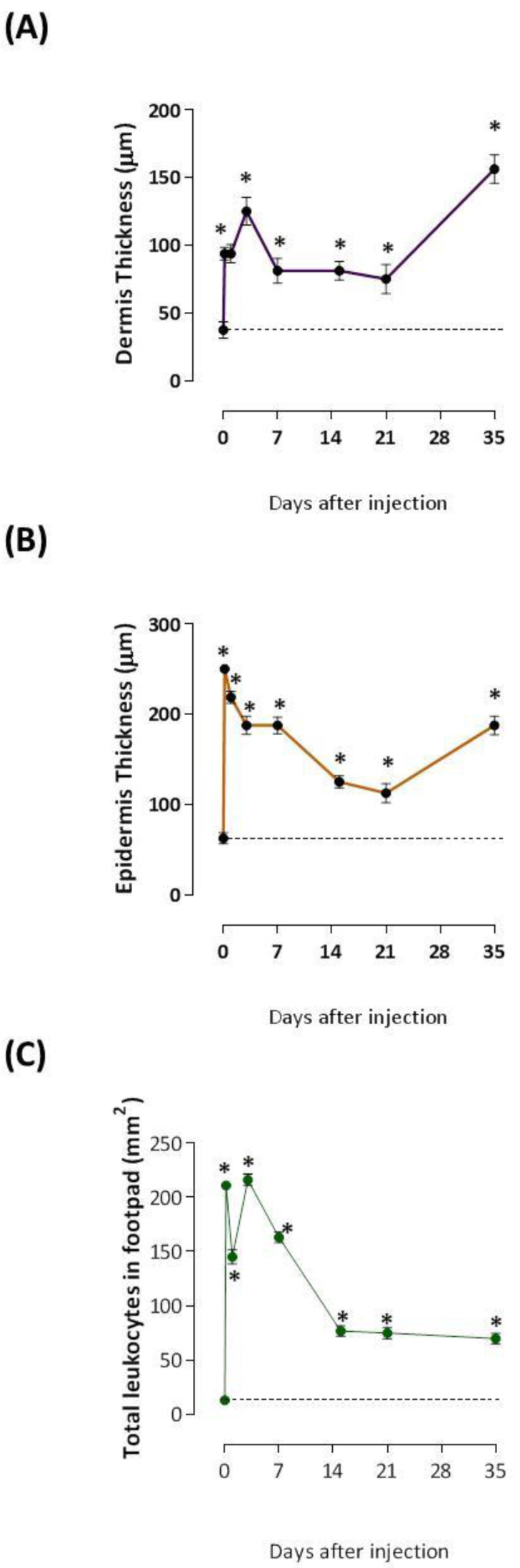
Quantitative analysis of structural changes in the epidermis and leukocyte influx. The paws of the mice from the control-group and the V*Tn* injected group collected after 4 and 24 h, 3, 7, 15, 21, and 35 days after injection were fixed, paraffinized, and sectioned at 5 μm for subsequent staining with H & E. Dermal (**A**) and epidermal (**B**) thicknesses were measured in photomicrographs of paw sections obtained after H & E staining by randomly selecting six regions. The total leukocytes (**C**) were counted by manual scoring on a 400 mm^2^ section per slide using Axio Vision Rel 4.8 software * *p* < 0.05 compared to the control-group at 0 h.

**Figure 3 ijms-24-08453-f003:**
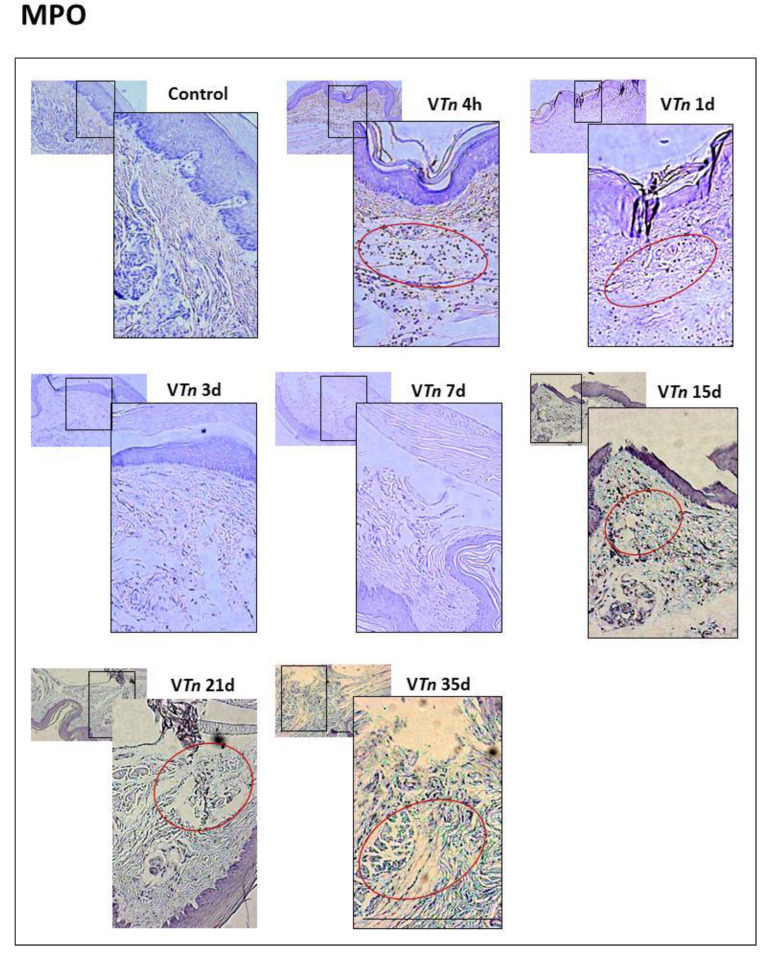
V*Tn*-induced injury shows persistent neutrophilia. The paws of the mice in the control-group and in the groups injected with V*Tn* were amputated, fixed, paraffinized, and sectioned at 5 μm. They were stained with anti-MPO antibodies for neutrophil detection which can be identified by the brown staining, and the group of MPO-positive neutrophils is indicated by red circles. All images were obtained with the Axio Imager A.1 microscope, Carl-Zeiss, Germany, coupled to the Zeiss AxioCam IcC1 camera in 10×, 40×, or 100× objectives using 1.6 optovar.

**Figure 4 ijms-24-08453-f004:**
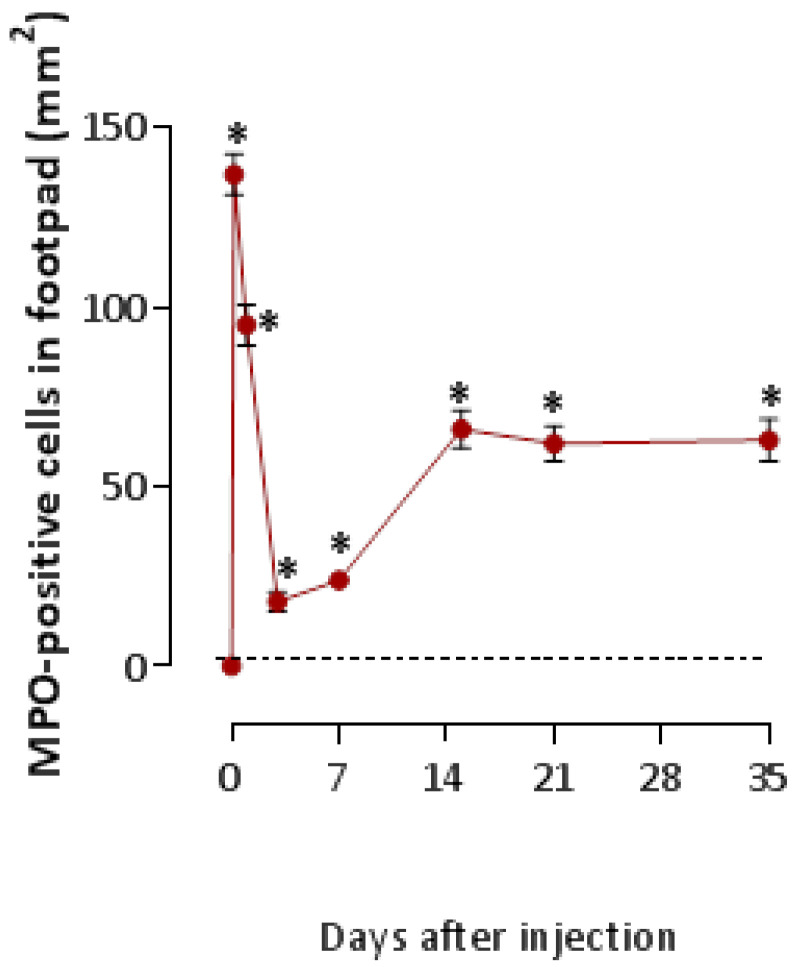
Quantitative analysis of MPO-positive neutrophils. The paws of the mice from the control-group and the V*Tn* injected group collected after 4 and 24 h, 3, 7, 15, 21, and 35 days after injection were fixed, paraffinized, and sectioned at 5 μm for subsequent staining with the anti-MPO antibodies. MPO-positive neutrophils were counted by manual scoring on a 400 mm^2^ section per slide using Axio Vision Rel 4.8 software * *p* < 0.05 compared to the control-group at 0 h.

**Figure 5 ijms-24-08453-f005:**
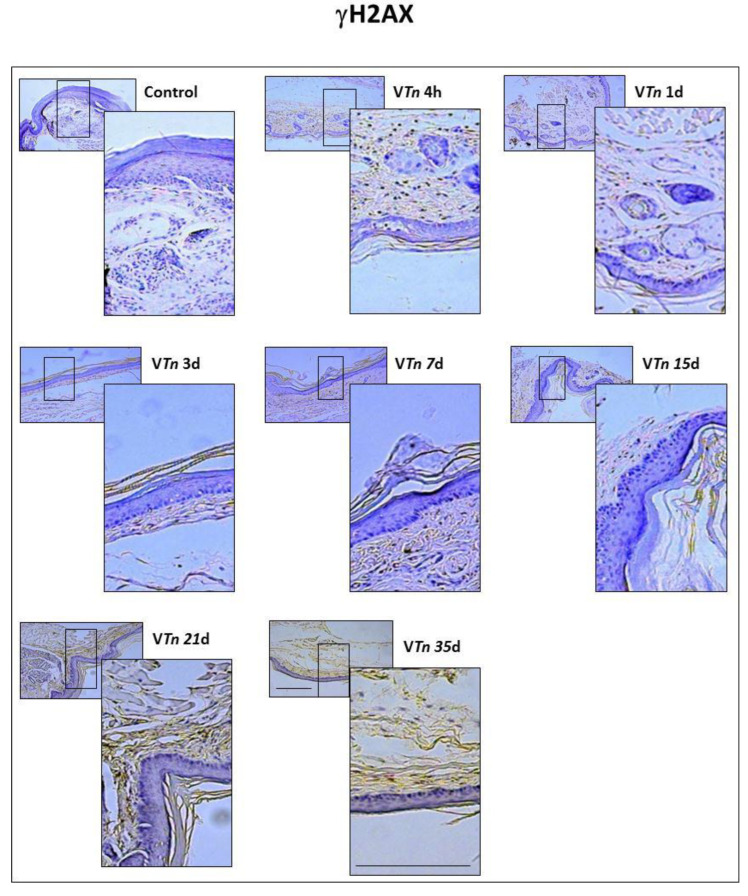
*T. nattereri* venom induces senescent lesions in the footpads of mice. The paws of mice from the control-group or those injected with V*Tn* collected after 4 and 24 h, 3 days, 7, 15, 21, and 35 days of the injection were processed and cut at 5 μm for staining with anti-mouse phosphohistone γH2AX antibodies. Sections were revealed by DAB chromogenic solution added with H_2_O_2_ for 5 min in the dark. Sections were counterstained with hematoxylin to visualize morphology. Foci of immunoperoxidase for γH2AX can be identified by the brown staining in either the nucleus or cytoplasm of all types of cells of the V*Tn*-injected group. The images were obtained with the Axio Imager A.1 microscope, Carl-Zeiss, Germany, coupled with the Zeiss AxioCam IcC1 camera in 10×, 40×, or 100× objectives using 1.6 optovar.

**Figure 6 ijms-24-08453-f006:**
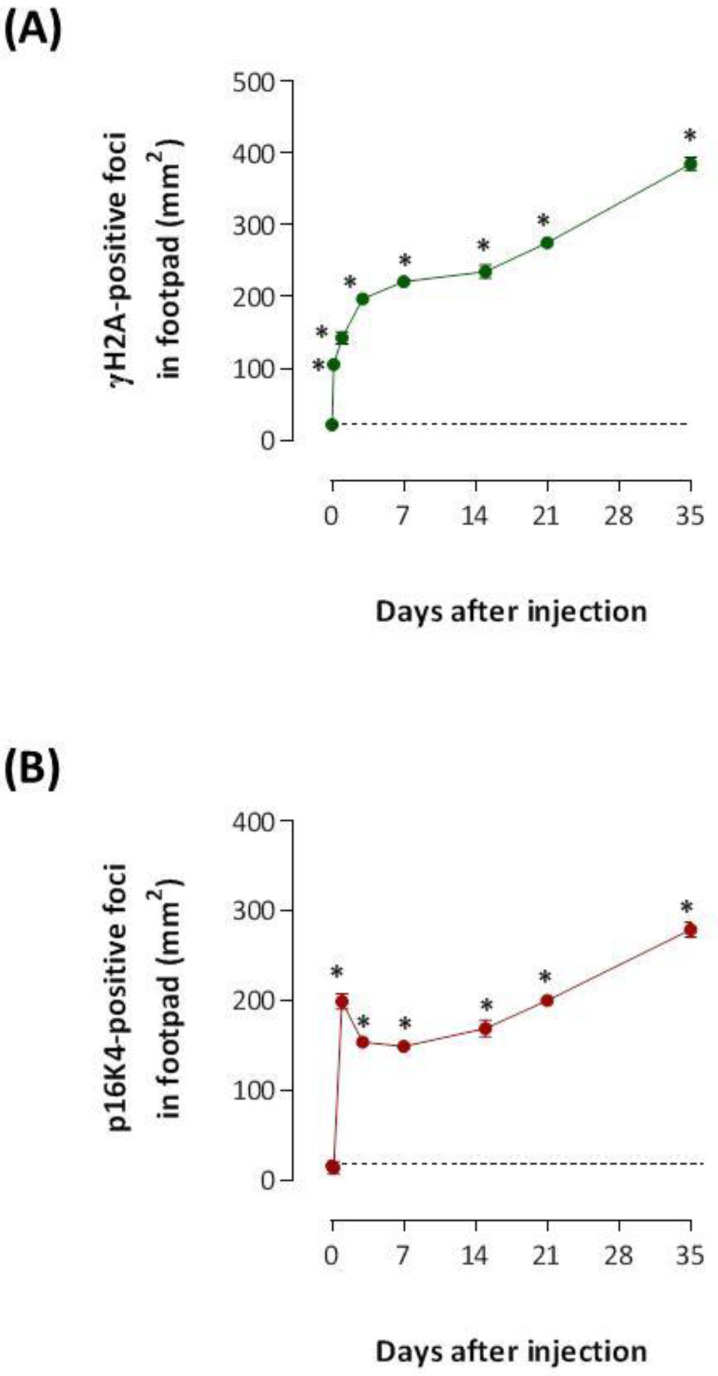
Quantitative analysis of senescence. The paws of the mice from the control-group and the V*Tn* injected group collected after 4 and 24 h, 3, 7, 15, 21, and 35 days after injection were fixed, paraffinized, and sectioned at 5 μm. Foci of immunoperoxidase staining for γH2AX (**A**) or p16^INK4a^ (**B**) in either the nucleus or cytoplasm were counted in tissue by manual scoring on a 400 mm^2^ section per slide and performed using Axio Vision Rel 4.8 software * *p* < 0.05 compared to the control-group at 0 h.

**Figure 7 ijms-24-08453-f007:**
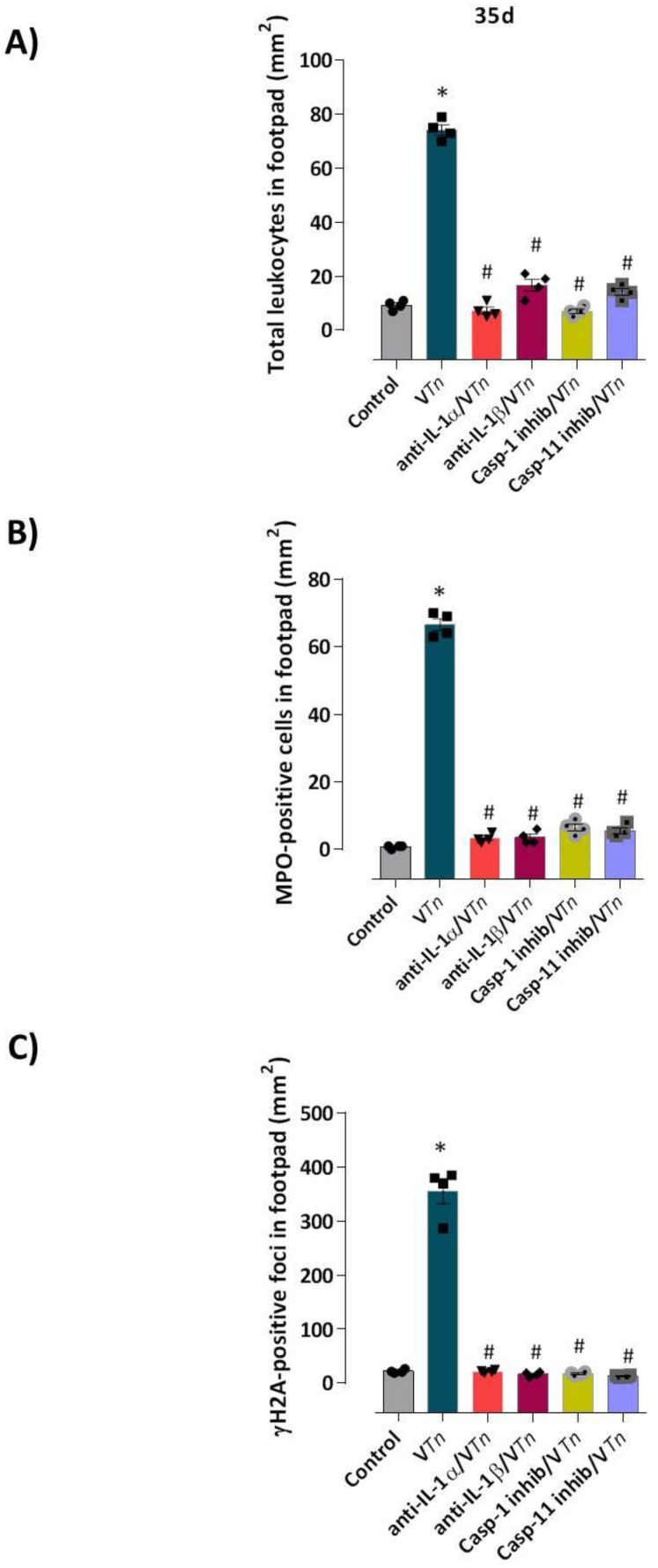
Quantitative analysis of senescence and neutrophilia mediated by IL-1α, IL-1β, and caspases-1 and -11. Mice were treated 30 min before V*Tn* injection with anti-IL-1α and anti-IL-β neutralizing antibodies, or caspase-1 (Ac YVAD-CMK) and caspase-11 inhibitors (Wedelolactone). The control-group received an equivalent volume of PBS. After 35 d, all groups of mice had paws amputated, fixed, paraffinized, and sectioned at 5 µm. The number of total leukocytes recruited to the paws was counted on slides stained with H & E (**A**), the number of neutrophils was counted on slides stained with the anti-MPO antibody (**B**), and senescence foci were counted on slides stained with the anti-γH2AX antibody (**C**) * *p* < 0.05 compared with the control-group at 0 h and # *p* < 0.05 compared with V*Tn*-group.

**Figure 8 ijms-24-08453-f008:**
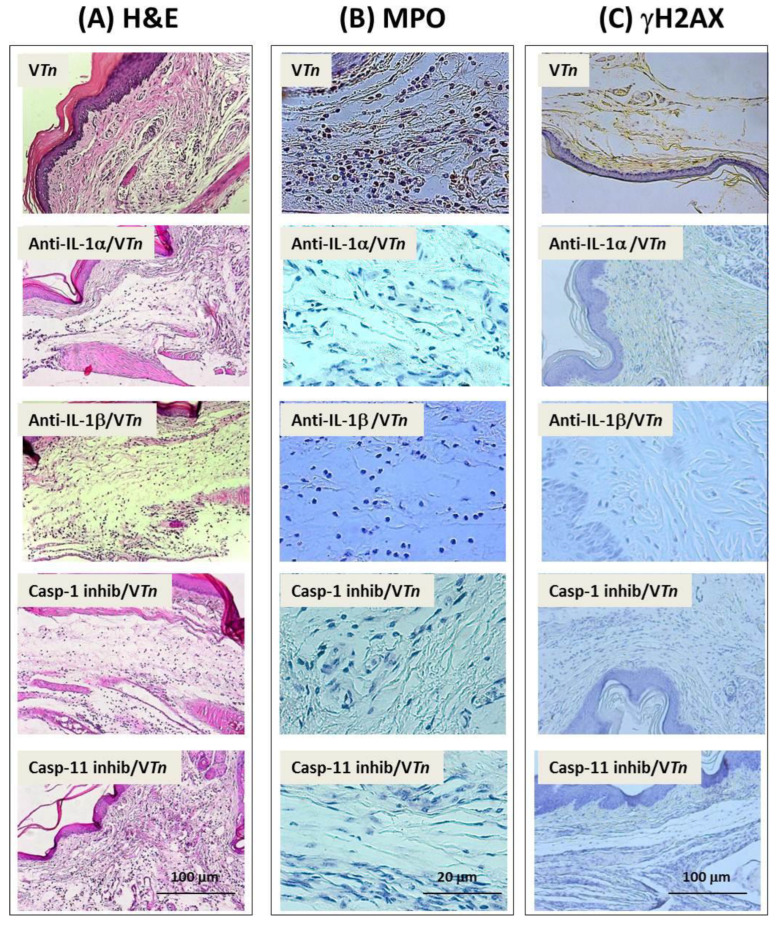
Senescence and neutrophilia induced by *T. nattereri* venom are mediated by IL-1α, IL-1β, and caspases-1 and -11. Groups of mice were treated 30 min before V*Tn* injection with anti-IL-1α and anti- IL-1β neutralizing antibodies, or caspase-1 inhibitor (Ac YVAD-CMK) and caspase-11 (Wedelolactone). After 35 d, the paw was amputated, fixed, paraffin waxed, and sectioned at 5 µm. Slides were stained with H & E (**A**), anti-MPO (**B**), and anti-γH2AX (**C**) antibodies. The images were obtained with the Axio Imager A.1 microscope, Carl-Zeiss, Germany, coupled to the Zeiss AxioCam IcC1 camera in 10×, 40×, or 100× objectives using 1.6 optovar.

**Figure 9 ijms-24-08453-f009:**
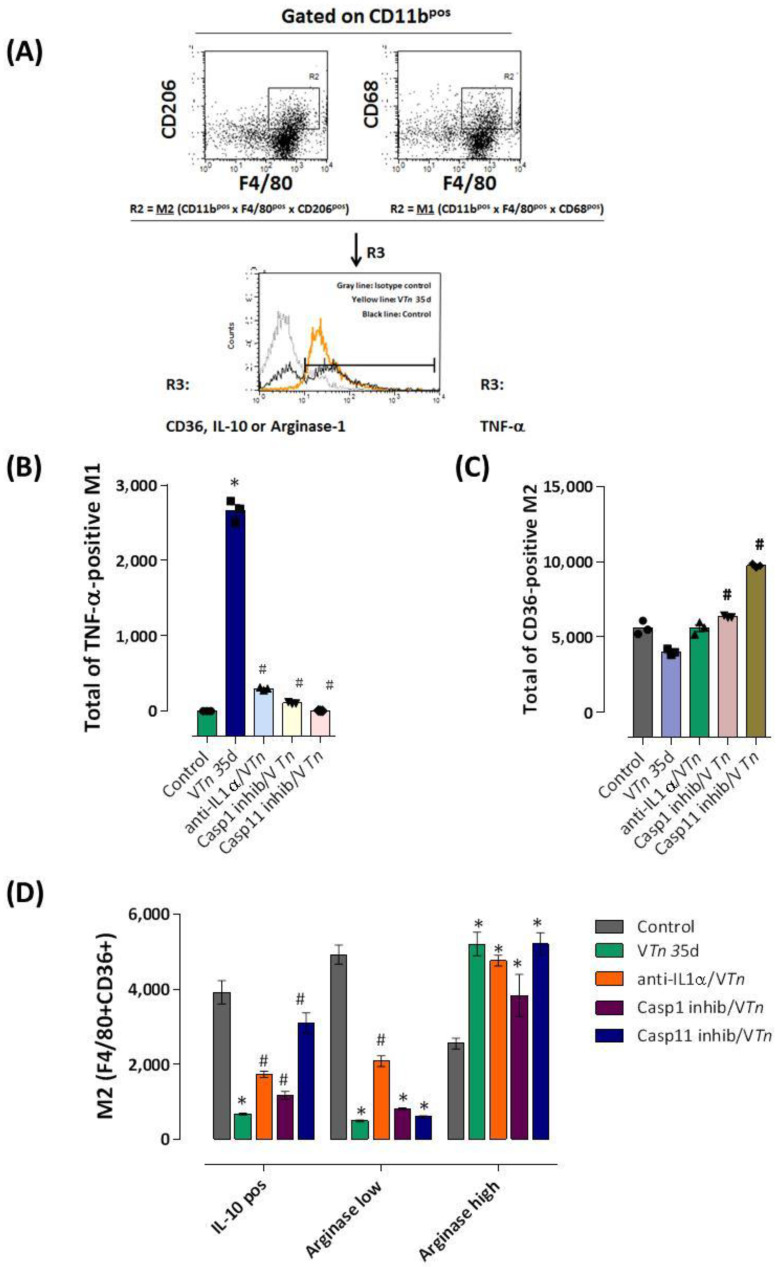
*T. nattereri* venom promotes an imbalance of M1/M2 macrophages dependent on IL-1α and caspase -1 and -11. Single-cell suspensions of foot pad tissue were prepared, counted, and analyzed by flow cytometry. (**A**) Representative dot plots show the frequency of CD11b^pos^ macrophages at the R1 gate, cells that were sequentially identified in the R2 gate as M1 (F4/80^pos^ CD68^pos^) or M2 (F4/80^pos^ CD206^pos^). The frequency of M1 macrophages staining positive for TNF-α, M2 staining positive for CD36, IL-10, or Arginase1 in the control-group, V*Tn* 35d, anti-IL1a/V*Tn*, casp-1 inhibitor /V*Tn*, and casp-11 inhibitor/V*Tn* were demonstrated in the R3, where the gray line indicates the isotype-control, the black line indicates the control-group, and the yellow line indicates TNF-α, CD36, IL-10, or Arginase-1 levels in these specific groups of mice. TNF-α-producing M1 (**B**), CD36-positive M2 (**C**), or IL-10-or arginase-1-producing M2 (**D**) were evaluated in groups of control mice, injected with V*Tn* or treated before injection of V*Tn* with IL-1α neutralizing antibodies or caspase-1 and -11 inhibitors. * *p* < 0.05 compared to control-group, and # *p* < 0.05 compared to V*Tn*-group.

## Data Availability

Not applicable.

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
