# Peer review of "Inflammasome Coordinates Senescent Chronic Wound Induced by Thalassophryne nattereri Venom"

_ijms, 2023, doi:10.3390/ijms24098453_

Round 1
Reviewer 1 Report
General Impression:
An interesting study, presented in a well-written manuscript, with a varied set data regarding the injury caused by the venom of Thalassophrine nattereri (VTn), a fish species, responsible for many accidents in the northeast/north regions of Brazil.
The authors investigated the participation of the cellular senescence and the inflammasome complex in the outcomes of chronic injury caused by the VTn, using classic tools such as flow cytometry, IHC and in vivo assays, including pharmacological inhibition, to investigate whether the injury caused promotes cellular stress, with morphological and transcriptional changes in the microenvironment, resulting in cell cycle arrest, and senescence.
The data presented support the initial hypothesis of the study, the conclusions are coherent with the data showed, and not much speculative. Nevertheless, I still have some "improvements and questions" to the authors – all easily implementable or easy to reply – before to recommend the publish this nice manuscript.
Questions, comments, and minor concerns:
1) Page 5, lines 244 – 246: Authors should further discuss data related to neutrophilia (15 to 21 days) and levels of MPO positive cells (up to 35 days). Are there similar reports in the literature about venoms from other fish species? If so, how diverse could VTn's mechanism of action be compared to other venoms already studied?
2) Page 7, figure 1A and B: Could the authors include in Figure 1, a photo of the paw of a control animal? I also recommend replacing the current photo of the treated animal (VTn 35 days) with another one of higher resolution.
3) Page 9, figure 2B: Could the authors provide images with higher resolution? It is so difficult to analyze the results based on current images. How was made the quantification of pINK4a expression performed in paw tissues injected with VTn?
4) Page 10, lines 306 – 309: Please check if the font size in this section is different from the rest of the text.
5) Page 14, figure 4: in the bottom of the figure 4, please remove "figure 4" keeping it only in the legend.
6) Page 15, conclusion: Although the text of the conclusion section is succinct and consistent with the data presented in the manuscript, it is certainly not based only on data from this study. Thus, it would be interesting if the text were revised, to make it clear to the reader what evidence, which supports the conclusion, was taken from previous studies by the group or even from the literature.
Author Response
IJMS (ISSN 1422-0067)
Section: Molecular Toxicology
Special Issue: Molecular Mechanisms of Animal Toxins, Venoms and Antivenoms
Manuscript ID: ijms-2228048R1
Inflammasome coordinates senescent chronic wound induced by Thalassophryne nattereri venom
Carla Lima, Aline Ingrid Andrade-Barros, Fabiana Franco Carvalho, Maria Alice Pimentel Falcao, Monica Lopes-Ferreira
Author's Reply to the Review Report
Reviewer 1
An interesting study, presented in a well-written manuscript, with a varied set data regarding the injury caused by the venom of Thalassophrine nattereri (VTn), a fish species, responsible for many accidents in the northeast/north regions of Brazil.
The authors investigated the participation of the cellular senescence and the inflammasome complex in the outcomes of chronic injury caused by the VTn, using classic tools such as flow cytometry, IHC and in vivo assays, including pharmacological inhibition, to investigate whether the injury caused promotes cellular stress, with morphological and transcriptional changes in the microenvironment, resulting in cell cycle arrest, and senescence.
The data presented support the initial hypothesis of the study, the conclusions are coherent with the data showed, and not much speculative. Nevertheless, I still have some "improvements and questions" to the authors – all easily implementable or easy to reply – before to recommend the publish this nice manuscript.
First of all, we appreciate you taking the time out to share your comments with us; we value and respect your opinion.
Questions, comments, and minor concerns:
1) Page 5, lines 244 – 246: Authors should further discuss data related to neutrophilia (15 to 21 days) and levels of MPO positive cells (up to 35 days).
Text describing the damaging power of neutrophils and their proteases including MPO was included in the manuscript.
Are there similar reports in the literature about venoms from other fish species? If so, how diverse could VTn's mechanism of action be compared to other venoms already studied?
As suggested we rewrite the paragraph that talks about the permanence of the neutrophilic response in the remodeling phase and its implication for chronic injury.
Many articles on the immunopathology of fish envenoming (Thalassophryne nattereri, Scorpaena plumieri, Potamotrygon gr. orbignyi and the Cathorops spixii) have been carried out by our group at Instituto Butantan, a traditional center for venom studies. Other groups have investigated the systemic effects of venoms mainly induced by Synanceia horrida (formerly known as S. trachynis) and S. verrucosa (Scorpaenoidei). With this, studies that describe the local injury caused by venomous fish are scarce.
2) Page 7, figure 1A and B: Could the authors include in Figure 1, a photo of the paw of a control animal? I also recommend replacing the current photo of the treated animal (VTn 35 days) with another one of higher resolution.
We included a photo of a normal paw and replaced the H&E of VTn 35d by another of better resolution.
3) Page 9, figure 2B: Could the authors provide images with higher resolution? It is so difficult to analyze the results based on current images.
Images containing the photos were separated and placed in a larger size
How was made the quantification of pINK4a expression performed in paw tissues injected with VTn?
The sentence in the MM has been rewritten.
Foci of immunoperoxidase staining for p16INK4a in either the nucleus or cytoplasm were counted in tissue by manual scoring on a 400 mm2 section per slide and performed using Axio Vision Rel 4.8 software. Discrete foci were counted as individual entities. Foci that appeared to overlap or merge were counted as one entity unless the overlap could be visibly distinguished.
4) Page 10, lines 306 – 309: Please check if the font size in this section is different from the rest of the text.
Manuscript formatting has been done and fonts are corrected
5) Page 14, figure 4: in the bottom of the figure 4, please remove "figure 4" keeping it only in the legend.
The numbering has been removed from the figures
6) Page 15, conclusion: Although the text of the conclusion section is succinct and consistent with the data presented in the manuscript, it is certainly not based only on data from this study. Thus, it would be interesting if the text were revised, to make it clear to the reader what evidence, which supports the conclusion, was taken from previous studies by the group or even from the literature.
The conclusion has been rewritten.
Reviewer 2 Report
Inflammasome coordinates senescent chronic wound induced by Thalassophryne nattereri venom
Summary: The injury response to Thalassophryne nattereri venom is evaluated.
Abstract
The abstract would be improved by detailing the common name of T. nattereri. This abstract would be improved by providing more details on the actual data collected and if significant.
Line 24- what treatments?
Introduction:
Line42-44, needs a reference
Line 60-61, this sentence needs to be reworded. “out by our group reproduced in mice…”
Line 66 The authors apparently change from VTn wound formation to ischemic injury and venous stasis without appropriate context.
Line 67 what cell activation?
Line 69 The necrotic injury…..combining with ischemic injury in the same paragraph will confuse the reader
Line 73: ….inhibit local 73 venom-induced toxic effects (e.g.???.)
Line 76-83 If this work has been done by the group, it needs to have references. It is unclear why the discussion of antibodies is in the introduction and not more connected with the current investigation.
The authors need better clarity on the definition of “microenvironment”
The manuscript would be improved by elaborating on cellular senescence and inflammasome signaling, since that is embedded in the language of the hypothesis.
Material and Methods
2.3
Line 121- The authors fail to provide information on how this VTn dose was selected. Why not use 0.33 ug/ul to describe VTn dose? What is the VTn dose in ug/mg? other chemical doses?
Line 127- how many groups? Mice per group? This needs to be added, not just located in the figure legend
Line 133-135 this sentence is unclear
2.4- Authors fail to adequately describe the experimental groups and controls. This should supplement the figure legends
Line 141- …”surface staining”……this language is unclear. Was 3% mouse serum used for blocking?
The authors fail to indicate if secondary antibody alone negative controls were used.
Line 156- What exactly are the negative controls?
2.5
Line 164 what groups?
2.6 – This section does not describe the negative controls such as incubation with secondary alone,
Line 172- “The 5um sections…” What sections, to which groups?
Line 179-184 is a run on sentence and needs to be separated.
Line 191 - this procedure is used repeatable and should have its own section to describe exactly how the quantification was performed.
Results:
General:It is unclear why the authors did not test the bioactivity of the venom over time in vitro. Is the venom still active after hours? Days? Are the results seen from the activity of the venom or secondary effects caused by the initial venom activity?
Figure 1A - The data provided would be improve by labeling various structure IDed in the discussion. In addition, providing a quantitative approach to the images such as measuring the thickness of the specific tissues layers. The magnification if the images are not adequate for the readers to draw the same conclusions. Having a higher magnification insert may achieve this.
If necrosis exist, the images do not seem to display destruction of tissue.
Line 233 It is unclear why the authors us “fold” for 1B and not in section 242-244.
Figure 1C - VTn 1d, 3d, 7d images are not clear. The figure would be improve by using arrow to indicate infiltrating cells.
Line 258 -This figure legend language makes the reader believe venom injections occurred over time. What color do the neutrophils stain? This will help the reader.
Figure 1B/D - The authors failed to provide std or std error in the graphs
Figure 2
Line 279 It is unclear why the term foci was used, compared to positive nuclei in line 274.
Figure 2 the low magnification images make it impossible for the reader to see “positive” cell examples.
Figure 2C – images presented are over exposed and not usable.
Figure B/D - not std or error presented in graphs.
The authors fail to show the control group data in the graph. Are the data compared to matched controls?
Figure 3
Figure 3D - the images presented are not clear. The authors need to indicate positive cells.
The study would be improved by showing the same data at an early timepoint rather than just the endpoint when the injury is mostly resolved.
The authors fail to quantify the changes in the treatment concentrations overtime. This would provide valuable information about the half life of the treatments. Do the circulating levels of the treatments fall back to physiological normal levels days or weeks before day 35?
Figure 4
The data would be better represented using conventional flow scatter plots or at least include this data in a supplemental document.
The authors need to show negative controls with secondary antibodies alone.
The authors need to specify the control in the figure legend
Author Response
IJMS (ISSN 1422-0067)
Section: Molecular Toxicology
Special Issue: Molecular Mechanisms of Animal Toxins, Venoms and Antivenoms
Manuscript ID: ijms-2228048R1
Inflammasome coordinates senescent chronic wound induced by Thalassophryne nattereri venom
Carla Lima, Aline Ingrid Andrade-Barros, Fabiana Franco Carvalho, Maria Alice Pimentel Falcao, Monica Lopes-Ferreira
Author's Reply to the Review Report
Reviewer 2
First of all, we appreciate you taking the time out to share your comments with us; we value and respect your opinion.
Abstract
The abstract would be improved by detailing the common name of T. nattereri. This abstract would be improved by providing more details on the actual data collected and if significant.
The Abstract has been rewritten
Line 24- what treatments?
The sentence in the Abstract has been rewritten: anti-IL-1α and anti-IL-b neutralizing antibodies and inhibitors of caspase-1 (Ac YVAD-CMK) and caspase-11 (Wedelolactone)
Introduction:
Line42-44, needs a reference
Line 60-61, this sentence needs to be reworded. “out by our group reproduced in mice…”
The sentence was rewritten.
Line 66 The authors apparently change from VTn wound formation to ischemic injury and venous stasis without appropriate context.
Line 67 what cell activation?
Line 69 The necrotic injury…..combining with ischemic injury in the same paragraph will confuse the reader
These sentences were rewritten.
Line 73: ….inhibit local 73 venom-induced toxic effects (e.g.???.)
The sentence was included: ….Venom-induced toxic effects in mice such as edema and nociception
Line 76-83 If this work has been done by the group, it needs to have references. It is unclear why the discussion of antibodies is in the introduction and not more connected with the current investigation.
The authors need better clarity on the definition of “microenvironment”
These sentences were rewritten.
The manuscript would be improved by elaborating on cellular senescence and inflammasome signaling, since that is embedded in the language of the hypothesis.
Information linking activation of the non-canonical inflammasome complex with senescence has been added to the Introduction
Material and Methods
2.3 -Line 121- The authors fail to provide information on how this VTn dose was selected. Why not use 0.33 ug/ul to describe VTn dose? What is the VTn dose in ug/mg? other chemical doses?
In the article published in 1998, we described that skin necrosis on the back of mice was induced by the injection of 100 ug of VTn in 50 uL of PBS. Concentration (weight/volume) = 2 mg/mL
In 2001, we described skeletal muscle necrosis and regeneration in mice injected intramuscularly in the right gastrocnemius muscle with 100 µg venom dissolved in 100 µL of PBS. Concentration (weight/volume) = 1 mg/mL
Here in this work we reproduced necrosis in the paw by injecting 10 ug in 30 uL of PBS in the right foot pad. Concentration (weight/volume) = 0.33 mg/mL
For each tissue an adequate volume was used and we used 6 times or 3 times less amount of VTn compared to the amounts previously used in those articles.
Lopes-Ferreira, M.; Bárbaro, K.C.; Cardoso, D.F.; Moura-Da-Silva, A.M.; Mota, I. Thalassophryne nattereri fish venom: biological and biochemical characterization and serum neutralization of its toxic activities. Toxicon, v. 36, p. 405-410, 1998.
Lopes-Ferreira, M.; Núñez, J.; Rucavado, A.; Farsky, S.H.P.; Lomonte, B.; Ângulo, Y.; Moura-Da-Silva, A.M.; Gutiérrez, J.M. Skeletal muscle necrosis and regeneration after injection of Thalassophryne nattereri (niquim) fish venom in mice. Int. J. Exp. Pathol. v. 82, p. 55-64, 2001.
Line 127- how many groups? Mice per group? This needs to be added, not just located in the figure legend
The sentence was included: ….Experiments using 3 to 5 mice per group were performed independently two times.
Line 133-135 this sentence is unclear
The sentence was corrected
2.4- Authors fail to adequately describe the experimental groups and controls. This should supplement the figure legends
Experimental groups were adequately described in the MM text and in the legends:
Control-group: mice injected only with 30 mL of sterile PBS
VTn-group: 10 mg of protein in a fixed volume of 30 mL of sterile PBS was subcutaneous injected into the plantar region of the right hind foot paw
anti-IL-1α/VTn-group: mice were treated with intraperitoneal (i.p.) injection of 200 mL of monoclonal mouse IgG1 anti-IL-1α at 2 mg/mL 30 min before application of VTn at 0.33 mg/mL
anti-IL-1b/VTn -group: mice were treated with intraperitoneal (i.p.) injection of 200 mL of polyclonal Goat IgG anti-IL-b at 2 mg/mL 30 min before application of VTn at 0.33 mg/mL
Casp-1 inhib/VTn -group: mice were treated with intraperitoneal (i.p.) injection of 200 mL of caspase-1 inhibitor at 1.76 mg/mL 30 min before application of VTn at 0.33 mg/mL
Casp-11 inhib/VTn -group: mice were treated with intraperitoneal (i.p.) injection of 200 mL of caspase-11 inhibitor at a dose of 80 mM 30 min before application of VTn at 0.33 mg/mL
Line 141- …”surface staining”……this language is unclear. Was 3% mouse serum used for blocking?
For labeling molecules exposed on the cell membrane, single-cell suspensions (1 x 106 cells in 100 mL) were first blocked by incubation for 30 min in a solution containing 3% normal mouse serum in ice and after incubated with specific anti-mouse Abs.
The authors fail to indicate if secondary antibody alone negative controls were used.
Line 156- What exactly are the negative controls?
Appropriate isotype controls were used as negative-controls.
2.5 Line 164 what groups?
The sentence was corrected
2.6 – This section does not describe the negative controls such as incubation with secondary alone
The sentence was corrected: Controls were performed by omitting primary antibodies from the immunohistochemistry procedure
Line 172- “The 5um sections…” What sections, to which groups?
For in histological analysis, group of mice (control, VTn or treated/VTn) sacrificed after 4 h, 1 d, 3 d, 7 d, 15 d, 21 d, and 35 d had their paws amputated and were fixed in a 10% formalin solution, then paraffinized, sectioned at 5 mm and then deparaffinized with xylene, rehydrated with alcohol and PBS (2x for 1 min) and fixed with 3% formaldehyde for 30 min at room temperature.
Line 179-184 is a run on sentence and needs to be separated.
Line 191 - this procedure is used repeatable and should have its own section to describe exactly how the quantification was performed.
The typical paragraphs of each procedure have been separated for greater clarity of the text
Results:
General:It is unclear why the authors did not test the bioactivity of the venom over time in vitro. Is the venom still active after hours? Days? Are the results seen from the activity of the venom or secondary effects caused by the initial venom activity?
The injection of VTn in the paw was used to mimic the accidents that occurred in humans, which present injuries mainly in the hands and feet. The use of the in vivo model recapitulates accidents in humans and makes it possible to see the interrelationship of the various cells involved in inflammation, chronic wound and senescence such as epithelial and endothelial cells, fibroblasts and keratinocytes and myeloid cell populations, such as macrophages and neutrophils.
Figure 1A - The data provided would be improve by labeling various structure IDed in the discussion. In addition, providing a quantitative approach to the images such as measuring the thickness of the specific tissues layers.
The magnification if the images are not adequate for the readers to draw the same conclusions. Having a higher magnification insert may achieve this.
Indications in the figures of the stratum corneum, dermis and epidermis were included.
Images containing the photos were separated and placed in a larger size.
Graphs showing the quantification of structural changes observed in slides stained with H&E have been added.
Epidermal and dermal thicknesses were measured according to in photomicrographs of skin sections obtained after hematoxylin and eosin staining by randomly selecting six regions using an image analysis system with a Zeiss AxioCam IcC1 camera in a 10x or 40x objectives using 1.6 optovar. The vertical thickness of the whole skin and the thickness of the epidermal layer were defined as the distance from the panniculus carnosus to the stratum corneum and as the distance from the basal layer to the stratum corneum, respectively.
Jeon, Y. J., Sah, S. K., Yang, H. S., Lee, J. H., Shin, J., & Kim, T. Y. (2017). Rhododendrin inhibits toll-like receptor-7-mediated psoriasis-like skin inflammation in mice. Experimental & molecular medicine, 49(6), e349. https://doi.org/10.1038/emm.2017.81
If necrosis exist, the images do not seem to display destruction of tissue.
The VTn-injected group exhibited paw necrosis with scab formation identified by disruption of tissue architecture with ghost cells and fewer discernible nuclei at 4 h, 1 d, 15 d, and 21 d compared to the control-group of mice. Histologically, the infiltration of neutrophil and also the formation of new dermis and epidermis under the necrotic tissue until 35 d were visible.
Line 233 It is unclear why the authors us “fold” for 1B and not in section 242-244.
With the quantification of the lesion the text was changed
Figure 1C - VTn 1d, 3d, 7d images are not clear. The figure would be improve by using arrow to indicate infiltrating cells.
Symbols were introduced in the figure to indicate the changes induced by the venom: red circle (leukocytes), green asterisk and thin green arrow (ghost cells), closed red arrow (scab formation)
Line 258 -This figure legend language makes the reader believe venom injections occurred over time. What color do the neutrophils stain? This will help the reader.
Subtitles have been rewritten
Figure 1B/D - The authors failed to provide std or std error in the graphs
Figure 2
Figure B/D - not std or error presented in graphs.
All graphs were created in Prisma and the mean are accompanied by the respective deviation, however some are minimal. Even so, the lines of the deviations were thickened and placed in black color for better visualization.
Line 279 It is unclear why the term foci was used, compared to positive nuclei in line 274.
Figure 2 the low magnification images make it impossible for the reader to see “positive” cell examples.
Foci indicate positive cytoplasmic or nuclear staining, and corrected text has been added to the manuscript
Figure 2C – images presented are over exposed and not usable.
Images containing the photos were separated and placed in a larger size
The authors fail to show the control group data in the graph. Are the data compared to matched controls?
Analyzes in the different periods of time were carried out in a paired groups: control x VT. Control-group did not present structural alterations in the dermis, epidermis, in the recruitment of leukocytes or senescence. The first graph point 0 h and the dotted line correspond to the control data.
Figure 3
Figure 3D - the images presented are not clear. The authors need to indicate positive cells.
The study would be improved by showing the same data at an early timepoint rather than just the endpoint when the injury is mostly resolved.
The authors fail to quantify the changes in the treatment concentrations overtime. This would provide valuable information about the half life of the treatments. Do the circulating levels of the treatments fall back to physiological normal levels days or weeks before day 35?
With the quantification of the lesion the text was changed
Figure 4
The data would be better represented using conventional flow scatter plots or at least include this data in a supplemental document.
The authors need to show negative controls with secondary antibodies alone.
The authors need to specify the control in the figure legend
A representative figure of the cytometric analysis strategy of macrophages M1 and M2 has been added

Round 2
Reviewer 2 Report
Authors,
The manuscript clarity has improved.
Introduction
Line 66-71 This sentence is a run on.
This correction is not located at line 73:
“Line 73: ….inhibit local 73 venom-induced toxic effects (e.g.???.)
The sentence was included: ….Venom-induced toxic effects in mice such as edema and nociception”
Material methods:
Clarity of presentation has improved
Results:
Figure 1&2 would be improved by providing a higher magnification insert, not just increaseing the size of the image.
Figure 4 &5 would be improved by providing a higher magnification insert, not just increasing the size of the image. Figure 5 images are not publication quality, and are clearly over exposed in the blue channel.
Authors fail to show negative control images in figures or provide that information in the response letter.
Author Response
IJMS (ISSN 1422-0067)
Manuscript ID: ijms-2228048 R2: Inflammasome coordinates senescent chronic wound induced by Thalassophryne nattereri venom
Authors: Carla Lima , Aline Ingrid Andrade-Barros, Fabiana Franco Carvalho, Maria Alice Pimentel Falcao, Monica Lopes-Ferreira
Reviewer 2
The manuscript clarity has improved.
R1: We have made all corrections as indicated and we incorporated all suggestions.
Introduction
Line 66-71 This sentence is a run on.
This correction is not located at line 73:
“Line 73: ….inhibit local 73 venom-induced toxic effects (e.g.???.)
The sentence was included: ….Venom-induced toxic effects in mice such as edema and nociception”
R2: The manuscript we received for corrections does not have line numbering. If any correction indicated by line was lost it was unintentional.
Material methods:
Clarity of presentation has improved
Results:
Figure 1&2 would be improved by providing a higher magnification insert, not just increaseing the size of the image.
Figure 4 &5 would be improved by providing a higher magnification insert, not just increasing the size of the image
R3: Fig. 1, 2, and 4 have been enhanced as indicated. Initially, the composite figures were divided. Now we place 2 figures of each slide of each analyzed group: a smaller one shows the slide of the entire paw and the other indicates the zoom of a detail that includes all the layers.
. Figure 5 images are not publication quality, and are clearly over exposed in the blue channel.
R4: Since senescence was confirmed by the gammaH2AX marker, the figure of p16INK4a expression in the paws with blues color has been removed and the graph relating to the positive-foci count has been maintained.
Authors fail to show negative control images in figures or provide that information in the response letter.
R5: We include in the manuscript a representative figure of how the cells were identified in the dot plot graph and the expression of markers as MFI graph. Representative intensity fluorescence line plot showing the frequency of M1 macrophages (CD11bpos F4/80pos CD68pos) staining positive for TNF-a, or M2 (CD11bpos F4/80pos CD206pos) staining positive for CD36, IL-10 or Arginase1 in control-group, VTn 35d, anti-IL1a/VTn, casp-1 inhibitor /VTn, casp-11 inhibitor/VTn were demonstrated. Examples of representative dot plots and MFI for TNF are below (pdf file included):

Round 3
Reviewer 2 Report
NA
Author Response
IJMS (ISSN 1422-0067)
Section: Molecular Toxicology
Special Issue: Molecular Mechanisms of Animal Toxins, Venoms and Antivenoms
Manuscript ID: ijms-2228048R3
Inflammasome coordinates senescent chronic wound induced by Thalassophryne nattereri venom
Authors: Carla Lima*, Aline Ingrid Andrade-Barros, Fabiana Franco Carvalho, Maria Alice Pimentel Falcao, Monica Lopes-Ferreira
Reviewer 2
We thank you for taking the time to evaluate our manuscript and we appreciate your suggestions for improving it. We corrected imperfections in English spelling and grammar and improved the way of describing the results. We believe that the selection of time periods used in the analysis of senescence and inflammation reflects the four phases of the regeneration process, which include acute inflammation, which lasts from 1 to 3 days; proliferation, which usually lasts from a few days to a month, and finally tissue remodeling which involves keratinocytes, fibroblasts, macrophages and endothelial cells to scar formation. We also believe that epigenetic markers and macrophage subpopulations analyzed are in line with the current literature.
Open Review
(x) I would not like to sign my review report
( ) I would like to sign my review report
Quality of English Language
( ) English very difficult to understand/incomprehensible
( ) Extensive editing of English language and style required
( ) Moderate English changes required
(x) English language and style are fine/minor spell check required
( ) I am not qualified to assess the quality of English in this paper
|
Yes |
Can be improved |
Must be improved |
Not applicable |
|
|
Does the introduction provide sufficient background and include all relevant references? |
(x) |
( ) |
( ) |
( ) |
|
Are all the cited references relevant to the research? |
(x) |
( ) |
( ) |
( ) |
|
Is the research design appropriate? |
( ) |
(x) |
( ) |
( ) |
|
Are the methods adequately described? |
(x) |
( ) |
( ) |
( ) |
|
Are the results clearly presented? |
( ) |
(x) |
( ) |
( ) |
|
Are the conclusions supported by the results? |
(x) |
( ) |
( ) |
( ) |
Comments and Suggestions for Authors: NA
Submission Date
02 February 2023
Date of this review
04 Apr 2023 18:10:07